# Direct observation of independently moving replisomes in *Escherichia coli*

Aleksandre Japaridze [1], Christos Gogou[1], Jacob W. J. Kerssemakers[1], Huyen My Nguyen [1] & Cees Dekker [1✉]

The replication and transfer of genomic material from a cell to its progeny are vital processes in all living systems. Here we visualize the process of chromosome replication in widened *E. coli* cells. Monitoring the replication of single chromosomes yields clear examples of replication bubbles that reveal that the two replisomes move independently from the origin to the terminus of replication along each of the two arms of the circular chromosome, providing direct support for the so-called train-track model, and against a factory model for replisomes. The origin of replication duplicates near midcell, initially splitting to random directions and subsequently towards the poles. The probability of successful segregation of chromosomes significantly decreases with increasing cell width, indicating that chromosome confinement by the cell boundary is an important driver of DNA segregation. Our findings resolve long standing questions in bacterial chromosome organization.

[1] Department of Bionanoscience, Kavli Institute of Nanoscience Delft, Delft University of Technology, Van der Maasweg 9, 2629 HZ Delft, The Netherlands. ✉email: c.dekker@tudelft.nl

DNA replication is vital for the reproduction of all organisms and accordingly has been subject to intense research. Despite numerous studies on model bacteria such as *Escherichia coli* (*E. coli*), a debate persists on the intracellular organization and mobility of replisomes during DNA replication. No active machinery has been identified to be responsible for the highly effective segregation of newly synthesized sister chromosomes, and the physical mechanisms at play have remained incompletely understood.

Replication in *E. coli* initiates at a single genomic site called *oriC*, where two multi-protein complexes, called replisomes, are assembled. The replisomes replicate the chromosomes bi-directionally away from *oriC*, and eventually, after duplicating the entire genome, meet again and terminate at specific Tus sites[1,2] in the *ter* region[3]. Two conflicting models have been proposed for the organization and positioning of the replisomes inside the cell (Fig. 1). The so-called 'factory model'[4] predicts that both replisomes are co-localized together at a joint position near cell center, where the maternal DNA is replicated into daughter replicates of the two chromosome arms. Consequently, the newly duplicated DNA is being pushed out toward the cell poles, resulting in proper chromosome segregation. By contrast, the "train track model"[5] predicts that the two replisomes move independently along the chromosome arms as they carry out their function. In the latter case, the replisomes are not necessarily positioned jointly at midcell, but they could of course very well still be observed there, especially near the start of the replication process where the two replisomes both start from *oriC*. Since the two replisomes would move independently, the train track model would require a separate mechanism to spatially drive chromosome segregation.

Various groups have used fluorescent microscopy techniques to study DNA replication and segregation in bacterial cells. Reyes-Lamothe et al.[6] observed replisome activity at different positions in the opposing cell halves throughout different stages of the *E. coli* cell cycle, thus arguing against the factory model with joint replisomes at a stationary site. However, in a more recent study, very different results were reported by Mangiameli

et al.[7], again based on fluorescently labelled replisomes in rod-shaped *E. coli*. Here, functional replisomes were observed to form a single focus proximal to the cell center throughout a major part of the cell cycle, suggesting the formation of a replisome factory.

Another major difference in literature possibly accounting for the conflicting replication models is the use of various *E. coli* strains and various growth conditions. Cass et al.[8] observed fluorescently labeled chromosomal loci to move toward midcell before segregation in MG1655 *E. coli* cells with doubling times <1 h. As segregation initiated, the dynamics of loci were universal and independent of genetic position, consistent with a factory model. It is known that the finer details of chromosome morphology is different between the MG1655 and AB1157 strains[9], while the short replication time also has an influence on the chromosome organization[10,11].

While the debate on the mobility of the replisomes proceeds, it is furthermore unclear whether and how the replisomes are involved in chromosome segregation, which occurs simultaneously with the replication. Various models were proposed to explain the segregation process. *E. coli* bacteria show an initial rapid separation of the newly synthesized origins of replication[12], similar to what happens in *Vibrio cholerae*[13] where the segregation is driven by the *ParABS* segregation system. Yet, no such active protein system has been found in *E. coli*. It has been debated whether cell wall confinement may play a role in the process of segregation. Early experiments[14] in spherical *E. coli* cells (lacking MreB filaments that maintain the rod-shape of the cell) indicated that the rod-shape was not essential for proper chromosome segregation. Based on polymer simulations, Jun and Mulder[15], however, proposed an entropy-driven segregation model, in which internal repulsive entropic forces act on chromosomes to spontaneously demix them in rod shape cells, as the demixed polymer state is thermodynamically favored over the mixed one. Youngren et al.[10] strengthened the idea that entropy demixing may play a role with experiments on fast-growing *E. coli* that showed that, throughout segregation, chromosomes spatially self-organize as branched-ring-polymers under rod-shape confinement, where minimization of the chromosome free

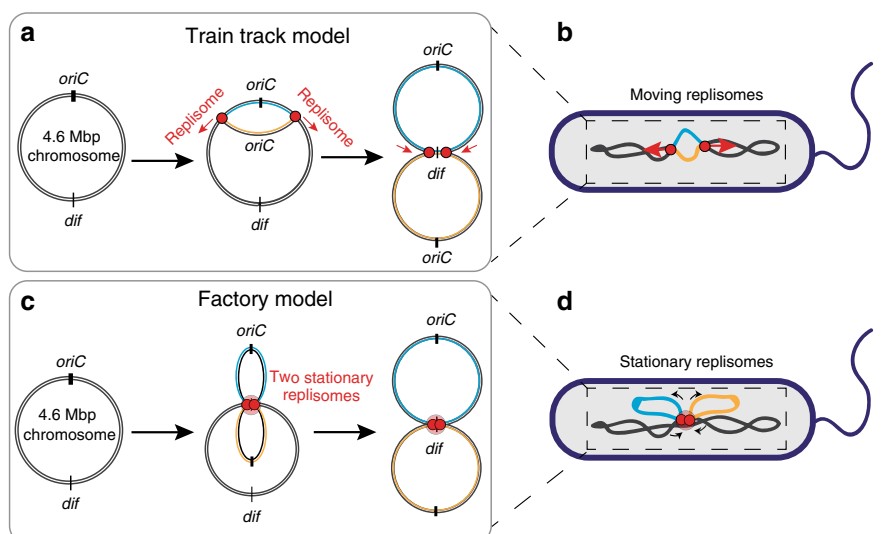

**Fig. 1 Schematic depiction of the "train track" (top) and factory (bottom) models for *E.coli* chromosome replication.** Train track model (**a**, **b**): according to the "train track" model, replisomes assemble at the origin of replication and then move independently along the chromosome arms (indicated with black arrows), replicating the DNA (replicated DNA depicted in orange and cyan colors). The replisomes then meet at the terminus region and thus finish replication. As sketched in **b**, one should accordingly expect to see two moving replisome foci upon tracking replisome foci in live *E.coli* cells. Factory model (**c**, **d**): by contrast, the factory model predicts that the two replisomes stay stationary after assembly at the origin of replication, while the DNA is being pulled through them (black arrows). As a consequence, replicated DNA (depicted in orange and cyan) is segregated by being pushed towards the cell poles. As sketched in **d**, labelled replisomes are thus predicted to form a single and stationary replication focus inside the cell.

energy can gradually drive newly synthesized sister chromosomes away from each other to opposing cell halves and consequently push non-replicated DNA toward midcell. In this picture, replisome positioning near midcell may result from the segregation process, rather than be its cause. Other groups have argued that mechanical strain rather than entropic forces are driving the chromosomal segregation[16,17].

Important questions on DNA replication thus remain unresolved, most importantly whether replisomes are separate and mobile or joint and static relative to the nucleoid, and what drives chromosomal segregation. There are various reasons why the answers to these questions varied in the studies reported so far. The aforementioned replisome visualization approaches differed in the types of fluorophores used, as well as in which components of the replisomes were tagged. In population-based assays, additional uncertainties arose when determining the protein positions due to their variability at various stages of cell cycle. For example, it is challenging to distinguish foci from newly initiated rounds of replication from single foci from ongoing rounds. Yet another technical challenge is the resolvability of independent replisomes due to optical resolution limits and the very compact state of the nucleoid in rod-shaped *E. coli*.

Here, we overcome these technical limitations by visualizing the process of replication of an individual chromosome in *E. coli* cells with an increased length or width. We achieve this by modifying the shape of cells using low concentrations of cell-wall attacking antibiotics, and by using a temperature-sensitive dnaC allele to generate cells with initially only one chromosome. In these temperature-sensitive (dnaCts) cells, DnaC protein is inactive above 40 °C, and, since DnaC protein is vital for loading the helicase (DnaB) onto the origin of replication[18], cells cannot initiate a new round of replication, while they will finish already initiated rounds. This enables us to synchronously initiate a single round of replication and eliminate the uncertainty whether observed foci in a single cell are from temporally overlapping replication rounds. Cells can maintain a single chromosome, elongate, and reach longer sizes compared to normal rod-shaped cells (Supplementary Fig. 1), with accordingly larger nucleoid sizes[19], overcoming the diffraction limit to resolve separate foci. We use well-established monomeric fluorophores that avoid the problem that replisome co-localization may result due to attracting forces between fluorescent tags. By initiating a single round of replication in the cells and tracking the number of replisome foci, we can determine whether the replisomes move independently or remain co-localized. This allows us to directly visualize the separation of two single replisome foci after replication initiation. We find that replisomes move away from each other, clearly moving along the nucleoid arms as they replicate the DNA. Furthermore, we, interestingly, observe that the cell's capacity to successfully segregate the replicated chromosomes towards daughter cells strongly depends on the cell width, indicating that entropic repulsion plays a role in DNA segregation.

## Results

**Replisomes load near the midcell and move away from each other.** To trace the position of the replisomes, we monitor a fluorophore-labelled key component of it, viz., the β-clamp (DnaN), which serves as a marker for the location of the replication fork[20]. The factory and train-track models predict clearly different results for the number and position of the replisome foci relative to the nucleoid (Fig. 1). According to the factory model, we would expect only a single replisome focus near midcell that is stationary throughout replication (Fig. 1c, d). By contrast, the train-track model predicts an initially formed single focus that separates into two foci with a twice lower fluorescent intensity,

that gradually move away from each other and away from the midcell (Fig. 1a, b). We performed experiments with temperature-sensitive (dnaCts) *E.coli* cells where both the β-clamps and the chromosome were fluorescently labelled (see MM). At permissive temperatures (40 °C), dnaCts cells maintain a single chromosome while growing longer in contour length (Supplementary Fig. 1, Supplementary movie 1). By placing these cells for a short time period (~10 min) at room temperature and then back to 40 °C, we could synchronously re-initiate replication in ~85% of all cells (Fig. 2, Supplementary Fig. 2).

As seen in Fig. 2a–c, cells placed at 40 °C maintained a single chromosome positioned at the center of the cell, as previously observed[19]. The intensity of the nucleoids was not homogeneous but showed a remarkable heterogeneity with a pronounced periodic-like structure, as reported by Fisher et al.[17]. High-DNA intensity regions were observed to occur semi-periodically along the nucleoid (Fig. 2d), where the number of the maxima was proportional to the length of the nucleoid. From the autocorrelation between the peaks, we estimate a periodicity of the undulation of $0.73 \pm 0.15\,\mu m$ (error bar denotes standard deviation (s.d.); $N = 30$, Fig. 2e). This observed periodicity of the nucleoids is nontrivial, especially given that the period appears to be independent of nucleoid length. Polymer modelling of chromosomes may elucidate this and it may be of interest to study whether this periodicity systematically varies as a function of cell length or cell cycle, opening up new possibilities for future experiments. Note that the average cell length as well as the nucleoid length at the point of initiation of replication are significantly expanded in these elongated cells at 40 °C (Fig. 2f, g). The average cell length and nucleoid length at 40 °C were $7.7 \pm 0.2$ and $3.3 \pm 0.1\,\mu m$, respectively, (errors are s.e.m.), and $4.7 \pm 0.05\,\mu m$ and $2.6 \pm 0.05\,\mu m$ ($\pm$ s.e.m.), respectively, for cells grown at 30 °C. At both temperatures, we found a linear dependence between the nucleoid length and the cell contour length, similar to other studies[19,21] (Fig. 2h, which was universal and not dependent on the strain type of *E.coli* used, nor of the labelling of the replisomes, Supplementary Fig. 3). Note that earlier work by our group showed that the size of individual nucleoids in elongated non-replicating cells reaches a saturation point for cells larger than ~15 μm,[19] but here we do not reach this saturation regime since the cells in our experiments were replicating and were shorter in length.

After this initial characterization of the nucleoids, we monitored the positioning of replisome foci relative to the nucleoid and cell contour. After initiating a single replication round, a single replisome focus was observed to form in the middle of the chromosomes, with a deviation from the cell center of, on average, 0.45 μm, which is ~5% of the average cell length (Fig. 2i, j). Shortly thereafter, i.e., in less than 15 minutes of initiation (which is 12% of the cell cycle time of 124 minutes in these conditions), the focus was seen to split into two foci in more than 70% of all cells ($N = 147$), each with an almost twice lower fluorescent intensity ($2.3 \pm 0.2$ lower intensity, $\pm$s.d., $N = 20$) (Fig. 2i, j, Supplementary movie 2). The splitting of the replisome foci was observed to occur independent of the number of chromosomes (Supplementary Fig. 4) and independent of temperature (as it was also observed in cells grown at 37 °C, Supplementary Fig. 5). After >30 minutes of replication initiation, the number of cells with two foci gradually started to decrease (Supplementary Fig. 6), indicating that the fluorophores gradually bleached (Supplementary Fig. 7) and/or the replisomes disassembled after finishing the replication.

These observations directly conform to the train track model, where the replisomes first assembled at oriC, showing only one focus at replication initiation since their inter-distance was not resolved due to the diffraction limit, whereupon two foci became

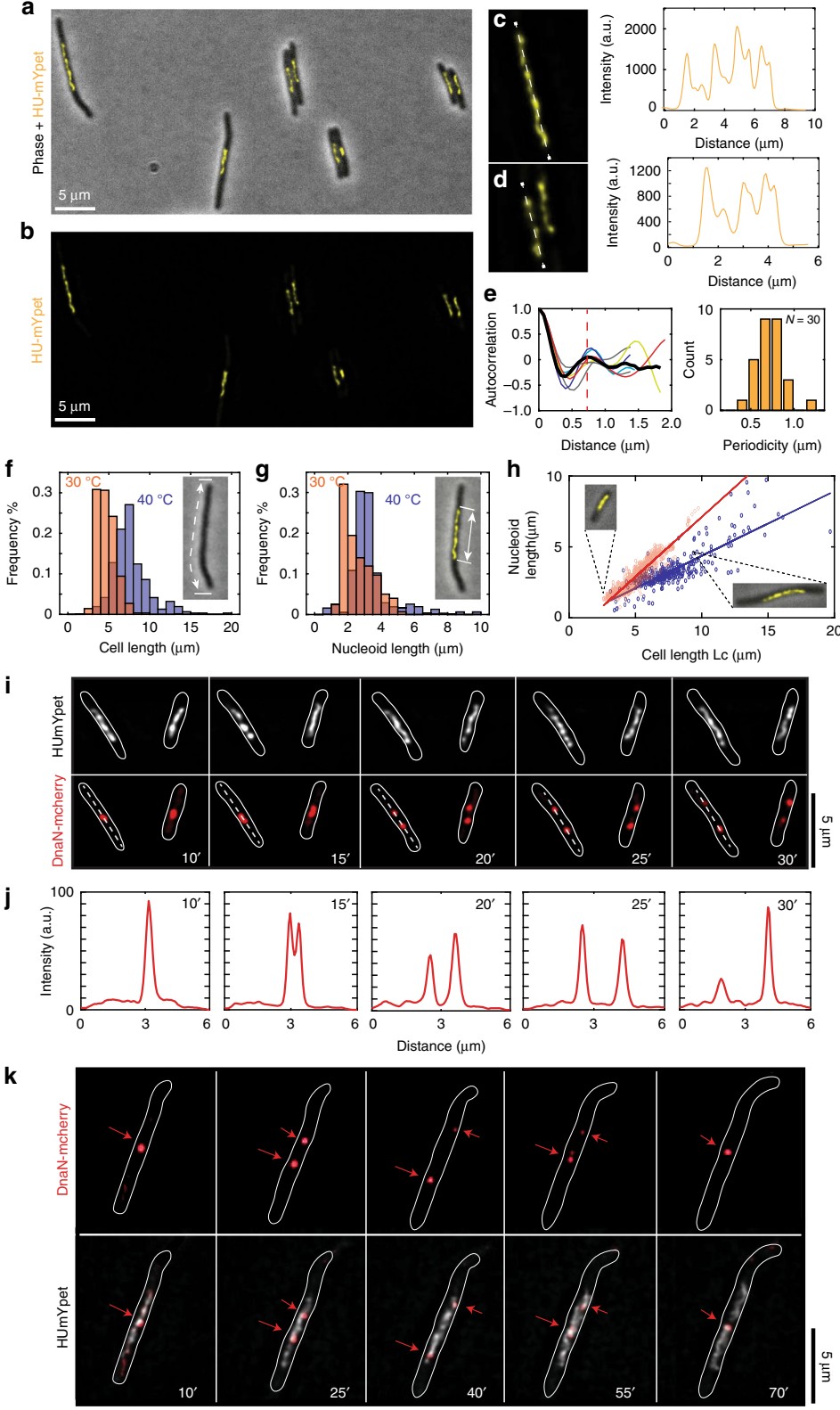

visible as the replisomes spatially separated, each replicating the material of an opposing chromosome arm in a different half of the cell length. After completing their function, the replisomes moved back towards each other (which was observed in ~25% of all replicating cells ($N = 84$), whereas the rest bleached during the measurement or had no or multiple replication bubbles) and after

finishing the replication (Fig. 2k) disassembled, reducing the observed number of foci per cell at replication termination.

**Independently moving replisomes form a replication bubble.** To visualize the replication bubble, we used the same dnaCts cells with the labelled DNA and replisome, but now added the MreB-

**Fig. 2 Imaging replication cycle in dnaCts _E. coli_ cells with replisome and chromosome labels. a** Typical image of elongated cells with an overlay of brightfield and HU-mYpet signals. **b** HU-mYpet signal of the same field of view as on **a. c, d** Zoomed images of nucleoids from image in panels b. with the corresponding fluorescent intensity plots along the nucleoids (shown with dotted white lines). **e** Autocorrelation function (left) of the fluorescent intensity profiles along the nucleoids ($N = 30$) (average autocorrelation is shown with thick black line and the average $0.73 \pm 0.15\,\mu m$ (s.d.) value is shown with red dotted line). The histogram (right) shows the mean values for the autocorrelation periodicity. Source data are provided as a Source data file. **f** Cell length distributions for _E. coli_ cells grown at 30 °C (orange bars, $N = 1476$) and at 40 °C (blue bars, $N = 314$). The inset shows a phase-contrast image of a cell with the cell length marked with dotted white line. **g** Nucleoid length distributions for the same _E.coli_ cells grown at 30 °C (orange bars, $N = 1476$) and at 40 °C (blue bars, $N = 314$) as on panel f. The inset shows the phase-contrast and HU-mYpet image of a cell with the nucleoid length marked with white arrows. **h** Chromosome length vsets show overlay of phase and mYersus cell length at the replication initiation point for cells grown at 30 °C (orange circles, $N = 1476$) and at 40 °C (blue circles, $N = 314$). Lines indicate linear fits, $L_{nucleoid} = 0.84 \cdot L_{cell} - 1.3\,\mu m$ (red) and $L_{nucleoid} = 0.45 \cdot L_{cell} - 0.2\,\mu m$ (blue). Insets show overlay of phase and mYpet signals of two typical cells. Source data are provided as a Source data file. **i** Time lapse images of replicating cells grown at 40 °C with chromosome (top: HU-mYpet) and replisome (bottom: DnaN-mCherry). Shortly after the replisomes assembled near the center of the cell, a single focus split into two foci, which gradually moved away from each other into opposite directions. **j** Intensity of the mCherry signal along the dotted lines for the cells shown in the **i**. Scale bars 2 μm. **k** Time lapse images of replicating cells with replisome (top: DnaN-mCherry) and chromosome (bottom: HU-mYpet) labels. Shortly after the replisomes assembled near the center of the cell, a single focus split into two foci, which gradually moved away from each other into opposite directions and then came back to the middle of the cell. Scale bar 5 μm.

inhibiting drug A22. Since MreB is essential for maintaining the rod shape of _E.coli_ cells[22], the addition of A22 (which inhibits the MreB polymerisation) results in cells that are significantly wider than normal rod-shaped cells[14,23,25]. Our earlier work showed that cells treated with A22 stay physiologically healthy and active, while the single nucleoid adopts a toroidal configuration within the larger available cell volume[25]. Importantly, these cells allow to follow the dynamics of the replisomes and the nucleoid upon re-initiating replication. To verify whether the replisomes indeed move along the chromosomal arms, we thus expanded the cells with the A22 drug and visualized the replication as it started within the toroidal-shaped nucleoid.

In about 80% of all replicating cells ($N = 167$), we could once again clearly observe a single replisome focus near the center of the cell, that quickly split into two lower intensity foci that had a $1.9 \pm 0.2$ lower intensity ($\pm$s.d. $N = 30$, Supplementary Fig. 8). The foci moved away from each other along the nucleoid arms while replicating the DNA. Interestingly, we observed that the position of the replisomes was closely associated with regions of higher DNA intensity: As the replisomes moved along the circular nucleoid, the high-intensity DNA regions also moved with them (Fig. 3b). From analyzing the correlation between the foci position and the clusters size (see M&M), we found that the single replisome foci (after splitting) were typically near DNA domains that contained up to 20% of the whole chromosome signal. These clusters contained on average twice more DNA than any random DNA cluster (Supplementary Fig. 9). These high-intensity regions represent spots where replisomes are actively duplicating new chromosome material and our experiments allowed us to track the process of DNA replication in real-time.

Interestingly, the data revealed an extended period where the newly replicated DNA stayed cohesed to the mother DNA before genuinely separating into a replication bubble. In other words, we observed that the sister nucleoids did not mutually segregate into two separate strands immediately after the initiation of DNA replication, but instead they stayed mutually cohesed along their length for some time (Fig. 3b, panel II and III). This in line with some earlier observations in rod-shaped cells[12,17,26]. This period of mutual cohesion of the replicated DNA was typically ~15 min. It was followed by a quick segregation of sister chromosomes, leading to the visualization of a clearly opened replication bubble as shown in Fig. 3b panel IV.

Despite having an increased cell size and a single nucleoid, the cells were capable of fully replicating the DNA and finishing the replication cycle, as seen in Fig. 3e, f. During replication, the nucleoids underwent large structural rearrangements, evolving from a toroid to a split toroid, to a figure-8 shape, to finally two

separated toroidal-shaped nucleoids in daughter cells. The widened cells thus could still replicate and segregate the DNA and fully divide the mother to daughter cells, showing that the MreB filaments and the associated rod shape of cells were neither necessary for proper cell division nor for chromosome segregation[14].

Summing up, the widening the _E. coli_ cells allowed to visualize the structural evolution of the nucleoid during replication at a level of detail that was inaccessible to earlier studies. We observed that, rather than pulling the DNA through a joint factory, the replisomes function independently and move along the chromosome arms while replicating the DNA (Fig. 3f, replisomes moved back towards each other in ~33% of all replicating cells; $N = 160$). The replicated DNA stayed cohesed to the mother DNA for ~15 min, after which a replication bubble opened and the replication proceeded until a figure 8 and finally two separate daughter chromosomes were formed.

**Chromosome segregation depends on the lateral cell wall confinement**. To better understand the nucleoid rearrangements, in particular the segregation of chromosomes, in the replication process during the cell cycle, we imaged an _E.coli_ strain with a HU-mYpet labelled chromosome and ori1 and ter3 FROS arrays[25] with widefield and SIM microscopy (Fig. 4; Supplementary Movies 3 and 4; Supplementary Fig. 10). Consistent with the experiments described above, about half of all replications (48% out of $N = 453$) displayed replication bubbles and figure-8 shapes, again confirming the train-track model for replisomes (Fig. 4b–d). We found that the Left-Ori-Right orientation of chromosomes that was reported earlier in rod-shaped _E. coli_ cells[21–24] was preserved in these widened cell too. We observed the origin of replication of single nucleoids to duplicate near the midcell location (Fig. 4e), but this positioning was not necessary for replication initiation, as cells with two chromosomes could also duplicate their origins distal from the midcell position (Supplementary Fig. 11). In contrast to earlier studies[10,27–30], we observed that shortly after splitting, the Ori foci moved into a random direction (Fig. 4f), towards a region within the cell cytosol that was not yet occupied by the nucleoid. This is a nontrivial finding, as previous studies suggested a fine-tuned mechanism that would drive the replicated ori's to move towards the cell poles[30]. In very wide cells, the two replicated Ori's could initially even orient along the short axis of the cell whereas only after some time, as the amount of replicated DNA locally increased, they would re-orient towards the long axis and the cell poles (Supplementary Fig. 12). This observation suggests a

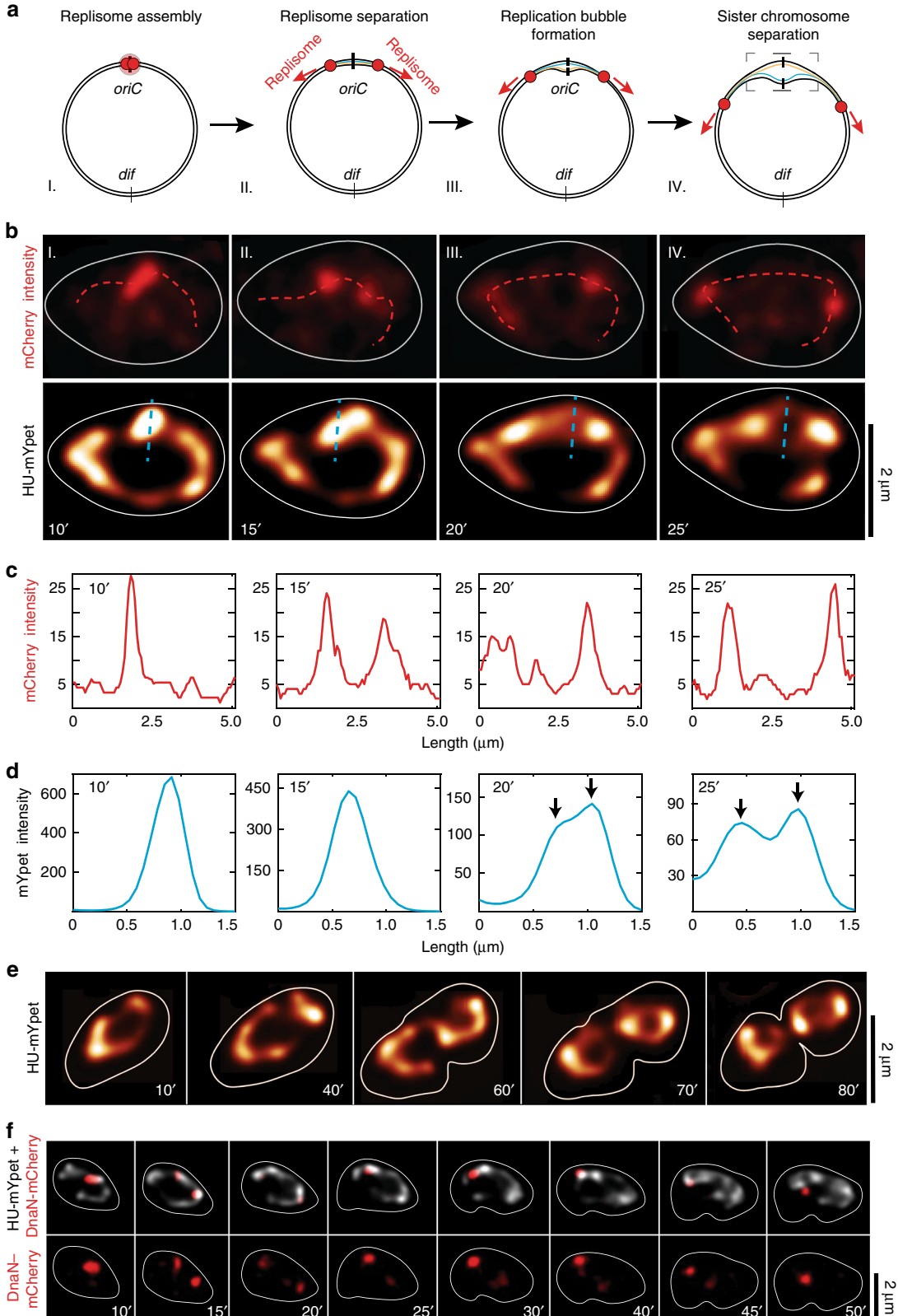

possible role for the spatial confinement of the cell wall in orienting the newly replicated nucleoids towards the cell poles.

When monitoring large cells during several hours of replication, we observed that they encountered difficulties in segregating their chromosomes (Fig. 4g and Supplementary Fig. 13). Quantitatively, a threefold increase in the cell area (from

$1.8\,\mu m^2$ to $5.2\,\mu m^2$, $N = 371$) at the moment of replication initiation decreased the probability of successful replication more than twofold (Fig. 4g). Interestingly, increasing the number of chromosomes in larger cells (by keeping the cells longer at room temperature), compensated for the loss of cell wall confinement and chromosomes did segregate properly (Supplementary Fig. 14),

**Fig. 3 Direct observation of replication bubble formation in widened cells. a** Schematics depicting the process of: I: replisome assembly II: replication fork splitting III: formation of the replication bubble and consecutively IV: sister chromosome separation and visualization formation of the replication bubble (dashed box). **b** Time lapse images of cell replication, depicting the same four steps as in the schematics shown in panel a. top: DnaN-mCherry signal and bottom: HU-mYpet signal shown in false color, the cell contour is shown as the continuous white line. **c** Intensity of the mCherry signal along the dotted lines on cells shown in the panel b top. The replisome foci quickly separate and then gradually move away from each other along the circular chromosome. **d** Intensity of the mYpet signal along the cyan dotted lines on cells shown in the panel b bottom. The intensity along the cross-section of the chromosome gradually widens, until the replication bubble forms and two sister chromosome regions are visible (indicated with black arrows). **e** Time lapse images of cell replication at longer timescales (HU-mYpet signal shown in false color). Cells do form a division septum at the geometric middle of the cell and eventually divide (80′). Note that the replicated chromosomes in daughter cells recover the toroidal shape of the chromosome, similarly to the mother cell. **f** Time-lapse images of replisomes moving along a circular chromosome and then coming back together. Top: overlay of HU-mYpet and DnaN-mCherry signal; bottom: DnaN-mcherry signal ($N = 53$). The cell contour is shown as the continuous white line.

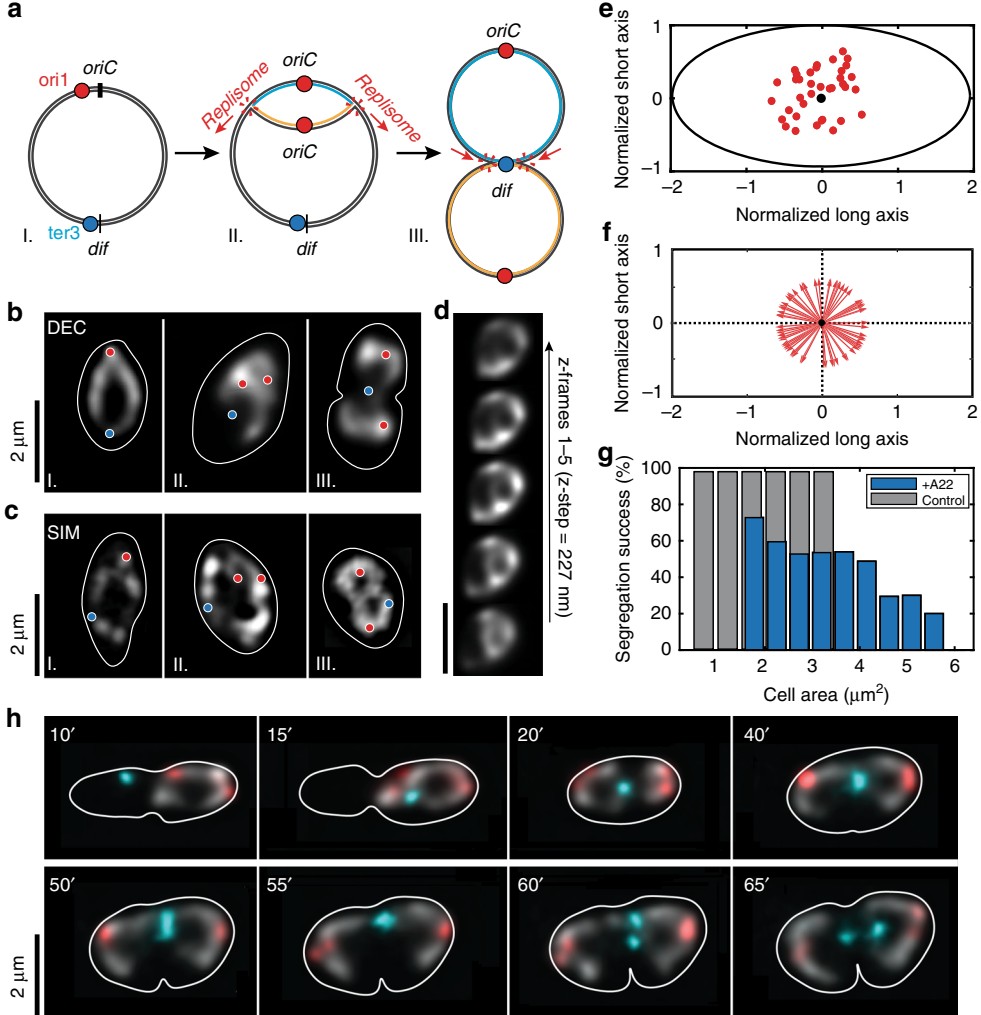

**Fig. 4 Time lapse images of expanded replicating cells with Ori, Ter and chromosome labels. a** Schematics depicting: I: labels on the circular chromosome, II: the formation of replication bubble, and III: figure-8 shapes by two replicated circular chromosomes that are still connected at the ter region near the *dif* site. **b, c** Deconvolved/2D SIM microscopy images of replicating cells depicting the same three shapes (circular chromosome, replication bubble and figure-8 shape) similar to the stages in the schematics shown in **a**. The circular chromosome is shown in greyscale, and ori1 and ter3 foci with red and cyan spots, respectively. The cell contour is shown as continuous white lines. **d** Deconvolved images of replication bubble in various height planes across the focal plane at the middle of the stack (z-step 227 nm). Scale bar 2 µm **e** Splitting positions of the Ori relative to the normalized cell size. The Ori focus preferentially splits near the geometric center (black circle) of the cells ($N = 38$). **f** After splitting, the Ori's move away from their splitting site in a random direction, without any preferential polar orientation as shown by point vectors emerging from the geometric center (black circle) depicting the Ori directionality ($N = 80$). **g** Successful chromosome segregation depends on the cell area at the start of replication, for rod-shaped (grey bars, $N = 139$) and A22 treated cells (blue bars, $N = 371$). **h** Time lapse images of replicating cells. The circular chromosome is shown in greyscale, Ori and Ter foci with red and cyan color, respectively. The cell contour is shown as continuous white line. The 10–20′ data show that the ter region moves towards the cell center where the future septum will form ($t = 40′$). The ter focus stays localized near the septum, until it duplicates ($t = 60′$) whereupon it quickly segregates to the sister halves ($t = 65′$).

similar to observations in earlier studies[14,23]. In rod-shaped cells grown at 30 °C, unsurprisingly, almost all cells ($N = 139$) successfully segregated their chromosomes (mean cell area $2.0 \pm 0.6 \, \mu m^2$; error is s.e.m.). These data also indicate that the confinement exerted by the cell wall directs the sister chromosome segregation in *E. coli* cells.

Shortly after the Ori foci splitting (typically around 15–20 min after replication initiation), the Ter foci moved towards the midcell location and stably positioned there for long periods of time (Fig. 4h, Supplementary movie 4). That the Ter foci position typically coincided with the midcell location suggests a spatiotemporal correlation between Ter and the divisome components[31]. Once the Ter focus duplicated, the foci quickly moved towards the respective cell halves enabling the septum to finish division of the cells and form two daughter cells (Fig. 4h). After division, both chromosomes recovered the toroidal topology, consistent with the notion that it is the physiologically active form of the *E. coli* chromosome[25]. As expected, inhibiting the supercoiling homeostasis[32] by perturbed the activity of TopoIV[33] and gyrases[34] upon administering Novobiocin drug reduced the capability of replicated chromosomes to timely separate[35] with up to 30% of cells ($N = 340$) ending up with cohesed chromosomes that were seen to get stuck at the division septum (Supplementary Figs. 15 and 16).

Our data thus show that the cell wall confinement acting on the chromosomes is important for the segregation of daughter chromosomes and that there is a fine-tuned temporal correlation between the DNA replication and cell division.

## Discussion

The process of DNA replication and its consecutive transfer from the mother cell to its progeny is one of the most intriguing processes in all living organisms. Surprisingly, even for the best studied organism, such as the *E. coli* bacterium, there is currently no unified view on the spatiotemporal control of DNA replication and segregation. Here, we addressed this by monitoring the replication and segregating of single chromosome in shape-modified cells. The use of a temperature-sensitive strain was advantageous for two reasons: first, it enabled us to synchronously initiate a single round of replication across the population of cells, and second, the nucleoids in these cells were elongated and thus permitted a higher spatial resolution (Fig. 2). Most importantly, we observed separate replisomes that independently moved from the origin to the terminus of replication along each of the two arms of the circular chromosome (Fig. 2k). We could visualize and study "text-book examples" of replication bubbles that provide direct evidence for the train track model describing the replication.

Some early studies on rod-shaped cells reported a single replisome focus near the midcell position pointing instead toward the factory model of DNA replication. We believe that the authors of these studies observed a single replisome focus because the nucleoids were highly confined which prevented the replisomes from moving further apart than the optical diffraction limit (~200 nm). Indeed, while in more recent work Mangiameli et al.[7] observed a single replication focus in ~80% of all cells, they, interestingly, also observed ~20% of foci to split into two foci for short periods of time, hinting that the replisomes could possibly spatially separate. Our experiments extend these studies and provide increased resolution to resolve that the factory model can be rigorously ruled out. A second major finding from our results is that chromosome segregation in slowly growing *E. coli* cells is found to be directed by the cell width. In rod-shaped cells, we saw that the cells did segregate the chromosomes properly for a variety of cell sizes. Upon using the A22 drug to widen the cells with single chromosomes, however, the segregation process

partially impaired. After replication initiation at OriC, newly synthesized DNA mass started to build up until the emerging two daughter chromosomes started to detach from each other, to, from there on, quickly segregate to form a replication bubble. The movement of the newly replicated DNA appeared to be driven by redistribution of DNA mass from strongly confined towards less confined space, which explains the initially random movement of the Ori's towards the available free space in the widened cells (Fig. 4f). When running into the confining cell boundary along the short axis, the segregating DNA material was re-distributed toward unoccupied space which was mainly available towards the cell poles (Supplementary Fig. 12). This also explains the observation that larger cells with single chromosomes had trouble segregating their chromosomes to the daughter cells (Fig. 4g, Supplementary Fig. 13). Interestingly, similar results were observed by the lab of Jeff Errington and colleagues in cell wall deficient (L-form) *Bacillus subtilis* cells, where forcing cells into narrow microfluidics channels increased the efficiency of successful chromosome segregation[36]. Our experimental data are supportive of the theoretical model by Jun and Mulder that predicted that entropy can be the driver for sister chromosome segregation[15] given a particular (rod-shape) confinement for given DNA concentrations[37]. Indeed, we observed that increasing the DNA amount inside a given confinement (by initiating multiple replication rounds in widened cells; Supplementary Fig. 14), one would recover a proper segregation of the chromosomes to the daughter cells.

Taken together, our results show that DNA replication in *E.coli* proceeds as predicted by the train track model of independently moving replisomes and that cell wall confinement plays an important role in segregating the nucleoids.

## Methods

**Strain construction.** All strains are derivatives of *E. coli* K12 AB1157 strain and were constructed by P1 transduction[38]. To construct strain AJ2818 (*dnaN-mCherry, hupA- mYPet:: frt, dnaC2 (ts):: aph frt*), mCherry–dnaN strain[39], a kind gift from Nynke Dekker, was transduced with P1 phage derived from strain FW1551[40] (*hupA-mYPet::aph frt*) for endogenous HU labelling. The resulting strain was cured of antibiotic resistance by flippase expressed from pCP20[41]. Then the strain was transduced with P1 phage FW1957[18] (*dnaC2(ts) ΔmdoB::aph:: frt*) to result in a DnaC temperature sensitivity. The final strain was selected for kanamycin resistance and temperature sensitivity and was subsequently cured of antibiotic resistance by flippase.

For experiments with Ori1/Ter3 labels, strain FW2179 (*ori1:: lacOx240::hygR, ter3::tetOx240::accC1 ΔgalK::tetR-mCerulean:: frt, ΔleuB::lacI-mCherry:: frt, hupA-mYPet:: frt, dnaC2 (ts):: aph frt*)[25], was used.

To generate MG1655 dnaCts and AB1157 dnaCts strains described in Supplementary Fig. 3, we first transduced the wildtype strains with *hupA-mYPet:: aph* from FW1551, then cured of kanamycin resistance using pCP20, and further transduced with *dnaC2 ΔmdoB::aph frt* from FW1957[25].

**Growth conditions.** For experiments with linear cells, we grew cells in liquid M9 minimum medium (Fluka Analytical) supplemented with 2 mM MgSO₄, 0.1 mM CaCl₂, 0.4% glycerol (Sigma-Aldrich), and 0.1% protein hydrolysate amicase (PHA) (Fluka Analytical) overnight at 30 °C to reach late exponential phase. On the day of the experiment, the overnight culture was refreshed (1:100 vol) for 2 h on fresh M9 medium at 30 °C. We then pipetted 1 μl culture onto a cover glass and immediately covered the cells with a flat agarose pad, containing the above composition of M9 medium as well as 3% agarose. The cover glass was then placed onto a baseplate and sealed with parafilm to prevent evaporation. The baseplate was placed onto the microscope inside a 40 °C incubator for 2 h to stop the cells from replicating and to let them grow longer. To reinitiate DNA replication, the baseplate was moved to room-temperature for 10 min before placing it back onto the microscope inside the 40 °C chamber for imaging.

To obtain circular chromosomes, we used the same protocol as described above, with minor changes: We grew cells in liquid M9 minimum medium (Fluka Analytical) supplemented with 2 mM MgSO₄, 0.1 mM CaCl₂, 0.4% glycerol (Sigma-Aldrich), and 0.01% PHA (Fluka Analytical) overnight at 30 °C to reach late exponential phase. On the day of the experiment, the overnight culture was refreshed (1:100 vol) for 2 h on fresh M9 minimal medium at 30 °C. We then pipetted 1 μl culture onto a cover glass and immediately covered the cells with a flat agarose pad, containing the above composition of M9 medium, A22 (final 3 μg/ml), as well as 3% agarose. The cover glass was then placed onto a baseplate and sealed

with parafilm to prevent evaporation. The baseplate was placed onto the microscope inside a 40 °C incubator for 2.5 h to stop the cells from replicating and to let them grow into round shapes. To reinitiate DNA replication, the baseplate was moved to room-temperature for 10 min before placing it back onto the microscope (inside 40 °C chamber) for imaging.

For treatment of replicating cells with Novobiocin, we first grew the cells in the presence of A22 as described above for 2.5 h to ensure they reach desired size and shape. Then we moved the baseplate to room-temperature for 10 min and afterwards added 10 µl of Novobiocin (~50 µg/ml final) to the agarose pad during replication initiation phase. Finally the cells were moved back to 40 °C chamber and imaged.

**Statistics and reproducibility.** All samples were repeated at least in biological duplicates.

Bacterial growth experiments in bulk. *E.coli* cells were grown on a clear-bottom 96-well plate (Nunc) with a final volume of 150 µl of the solution in each well. The plates were loaded into an Infinite 200Pro fluorescence plate reader (Tecan, Männedorf, Switzerland) and incubated at 30 °C in the presence of various concentrations of Novobiocin drug (25, 50, and 100 µg/ml). Samples were shaken with orbital agitation (2.5 mm amplitude) for a period of ~12 h. The cell density was measured at 600 nm with 15 min intervals, measured in biological triplicates.

**Fluorescence imaging.** Wide-field Z scans were carried out using a Nikon Ti-E microscope with a 100X CFI Plan Apo Lambda Oil objective with an NA of 1.45. The microscope was enclosed by a custom-made chamber that was pre-heated overnight and kept at 40 °C. mCerulean was excited by SpectraX LED (Lumencor) $\lambda_{ex} = 430$-$450$ through a CFP filter cube ($\lambda_{ex}/\lambda_{em} = 426$–$446/455/460$–$500$ nm). mYPet signal was excited by SpectraX LED $\lambda_{ex} = 510/25$ nm through a triple bandpass filter $\lambda_{em} = 465/25$–$545/30$–$630/60$ nm. mCherry signals was excited by SpectraX LED $\lambda_{ex} = 575/25$ through the same triple bandpass filter. Fluorescent signals were captured by Andor Zyla USB3.0 CMOS Camera. For each channel, between 3 and 11 slices were taken with a vertical step size of 227 nm (in total up to 2.3 µm).

Structured Illumination Microscopy imaging was carried out using a Nikon Ti-E microscope and a SIM module. A 100X CFI Apo Oil objective with an NA of 1.49 was used. Samples were illuminated with 515 nm laser line and a Nikon YFP SIM filter cube. mYPet, mCerulean, and mCherry signals of the same sample were also captured through wide-field imaging using a Nikon-Intensilight lamp. Filter cubes used for the wide-field imaging corresponding to the SIM images were CFP filters ($\lambda_{ex}/\lambda_{bs}/\lambda_{em} = 426$–$446/455/460$–$500$ nm), YFP filters ($\lambda_{ex}/\lambda_{bs}/\lambda_{em} = 490$–$510/515/520$–$550$ nm), and RFP filters ($\lambda_{ex}/\lambda_{bs}/\lambda_{em} = 540$-$580 / 585 / 592$–$668$ nm). Up to 19 slices were taken with a vertical step size of 100 nm (in total 1.8 µm). SIM image reconstruction was done by using NIS-Elements (version 4.51) software. During image reconstruction, special care was taken to use the recommended parameters to avoid reconstruction artefacts. Furthermore, care was taken to check for photobleaching during image acquisition (which was negligible), to minimize drift during imaging, and to avoid artifactual signatures in the Fourier transforms of the reconstructed images[42] (Supplementary Fig. 17).

**Deconvolution.** Image stacks of 3–19 slices of Z stack in wide-field imaging were deconvolved using the Huygens Professional deconvolution software (Scientific Volume Imaging, Hilversum, The Netherlands), using an iterative Classic Maximum Likelihood Estimate (CMLE) algorithm with a point spread function (PSF) experimentally measured using 200 nm multicolor Tetrabeads (Invitrogen). The PSF of the single-frame non-deconvolved widefield images had a FWHM of 350 nm horizontally and 800 nm vertically. Deconvolution, to a great extent, reduced the out-of-focus noise in the images, which also led to an improvement in lateral resolution.

**Reporting summary.** Further information on research design is available in the Nature Research Reporting Summary linked to this article.

## Data availability

The source data underlying Fig. 2e, h are provided as a Source data file. The raw microscopy data supporting the findings of this manuscript (Figs. 2–4, and Supplementary Figs. 1, 2, 5, 6, 7, 10, 11, 13, 14, 16 and 17) are freely available for download from 4TU.ResearchData repository [https://data.4tu.nl/repository/uuid:23de478d-8d08-4149-abd6-3398cbe14847]. All data is available from the authors upon reasonable request. Source data are provided with this paper.

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

## Acknowledgements
We thank Nynke Dekker and Sumit Deb Roy for the kind gift of replisome-labelled *E.coli* strain, and them as well as David Sherratt for discussions. The work was supported by ERC Advanced Grant SynDiv (no. 669598) to C.D., and by the Netherlands Organisation for Scientific Research (NWO/OCW), as part of the NanoFront and BaSyC programs. A.J. acknowledges support by the Swiss National Science Foundation (Grants P2ELP2_168554 and P300P2_177768).

## Author contributions
A.J. and C.D. conceived and designed the project. A.J. and H.M.N. constructed the bacterial strains. A.J., C.G., and H.M.N. did the microscopy experiments. J.K. led image analyses. All authors wrote the paper. C.D. supervised the project.

## Competing interests
The authors declare no competing interests.
