## [Peer Review File · Nature Communications]

Reviewers' comments:

Reviewer #1 (Remarks to the Author):

This is a beautiful and important study that should and could have appeared ten years in the bacterial chromosome community. The authors employed classic *ts dnaC* mutant to synchronize replication initiation. Furthermore, by treating *E. coli* with a low dosage of A22, cells lose their rod shape and become round, widening their width. This allowed them to clearly visualize the morphology of the fluorescently labeled chromosome and simultaneously track progression of DNA replication from a single intact chromosome using fluorescence microscopy. The end result of this relatively straightforward approach is remarkable. Before initiation, we see a torus morphology of the circular chromosome. Once replication starts, two replisomes split, unambiguously refuting the factory model. Newly replicated DNA stay cohesed for several minutes but eventually they separate. Near the end of replication, we get to see the figure-8 shape and their separation into two torus. The sequence of these images are worth a figure in a biology textbook, and it is surprising why this type of clear visualization took so long.

Perhaps the most striking conclusion of this study is the robustness of chromosome segregation and the role of cell shape in providing the directionality to chromosome segregation. This is similar to the observation by Jeff Errington's group for L-form bacteria. They observed severe defects in chromosome segregation in round cells without cell walls (L-form). However, when the L-form cells were forced to grow in narrow microfluidic channels so that the cells are elongated, chromosome segregation was restored. Errington presented the results in a number of talks at meetings, but somehow I cannot find the reference. As the authors show that spatial positioning of *ori* (by MukBEF; Ref 29) is not required for chromosome segregation, these observations for the first time provide strong evidence *in vivo* for the entropy-driven segregation models and the role of spatial confinement that has been around for over a decade.

Overall, the message of the study is simple, clear, and important and should be of very broad interest. The writing is also admirable. In conclusion, I recommend the paper for Nature Communications wholeheartedly.

Minor comments:

Typo L194: bible -> bubble

Reviewer #2 (Remarks to the Author):

Major Comments

The main aim of the study is to discern between two models for replication: the train track and the replication factory models. For this, the authors look at replication dynamics in a strain with a *dnaC^{ts}* allele to synchronize cells and therefore follow replication of a single chromosome. In addition, in a second set of experiments, they use conditions under which cells are round-shaped to increase the resolution of detection of the replisomes. The authors set the question as a rebuttal of one of the two existing models for replication. Their results clearly indicate that replisomes can exist as separate entities. This was observed before by other groups (Grossmann, Rudner, Wiggins, Sogaard-Andersen; Berkmen 2006; Wang, 2013; Mangiameli, 2016; Harms, 2013). But I am not convinced that they provide enough evidence for the tracking model.

My first concern is with their observation of nucleoid morphology under their conditions. Nucleoids as observed by HU-mYpet labeling looks abnormal: they occupies only a fraction of the cell width and appear abnormally condensed. These abnormalities are usually observed in cells suffering from phototoxic effects (eg. DNA damage) or after deconvolution artifacts.

A first conclusion of the ms is that nucleoids display helical patterns. How many cells display this pattern? Other papers claimed similar things (eg. this has been observed before for *Bacillus* (see Ben Yehuda, PNAS 2008)) but this was later disproved by other studies (e.g. Marko tested the hypothesis of helicity in their Mol Micro paper in 2012, see Yazdi et al) but found no evidence for this. I would thus be aware of any interpretation regarding nucleoid helicity, even more when the nucleoid images appear artifactual (see comment above).

One main result is that 1 replisome focus appears at midcell, then splits and moves away. This is interpreted to be in support of the train track model. But in this model one would also expect the replisomes to come back together and fuse at midcell as they turn to the replication of the terminus. Why is this not observed?

The experiment presented in Fig 1 has already been realized in Mangiameli et al (Plos Genetics 2016) under seemingly very similar conditions. In that case, the main conclusion was that replisomes remain close to midcell. Why are the results so different?

The authors explain that the number of cells with 2 replisome foci decreases (drastically) from 70 to 35% after 30' because of photobleaching or replisome disassembly. But photobleaching should be a gradual effect, not drastic (this can be actually measured by following the focus intensity over time...). But why would the replisomes disassemble before they replicate the *ter*?? In whatever case it seems to me that they must fuse before. Otherwise, it could be that disassembly is due to phototoxic effects.

The experiment in Figure 2 k,l is key to the author's claim that they observe replisome tracking. However a single example is shown. Authors should perform analysis based on a statistically relevant number of cells where they show single-cell traces of the distance between replisomes as a function of time. This is essential to support their claims. Statistics of number of replication spots from distinct cells at different time intervals (Figure S3) do not provide any support for a tracking model as both tracking and factory models would yield similar results.

In Fig. 3 the authors use A22 to induce changes in cell morphology. These lead to expansion of the cell volume and the decompaction of the nucleoid with respect to the conformation of the nucleoid in normal cells. It is an interesting method but I wonder whether the behaviour of replisomes and segregation is relevant to wildtype conditions given these large changes in chromosome compaction and organization.

The data in Fig. 3b are very unclear. The authors seem to show DNA images annotated with a circle that seems to indicate the positions of replisomes. Are the positions of these circles based on images of DnaN as in Figure 2? If it is the case, please show raw data and merged images. Otherwise, I just don't understand how they can get the positions or replisome foci based purely on DNA signal.

I cannot really understand how the authors can correlate the schemes in 3a to the images/traces in 3c-d. In the schemes they see the appearance of a replication bubble that increases in size as replication progresses. However, in the images below we observe that replisomes do not appear at the regions of high DNA density but more often at their borders (panels I-III) or even far away (panel IV). Also, it is not clear in panel IV what this high density regions are as they do not seem to be the regions where DNA is being replicated.

Then the authors claim they see sister chromatid cohesion, however they just show an image where they speculate a shade in the fluorescence signal may correspond to the replication bubble and where more intense DNA regions may correspond to cohesed sisters. But this is not evidence for sister chromatic cohesion. The authors should provide experimental proof for this claim.

The authors show in 3e that A22-treated cells are able to complete chromosome replication and segregation. This is good to see. But it would be important to show where the replisomes are during this process. Why don't they show the replisomes images? You would expect the replisomes coming together to the link in the figure-8. This could be an important support of their results.

The authors then use a strain with a FROS at oriC and a second close to ter and image how these FROS organize in cells treated with A22. A main conclusion, that is different from previous studies, is that newly replicated origins move away from each other isotropically. But the experiments in this paper are performed with this widened cells where cytoplasm volume is much larger, therefore I do not see the relevance to wildtype conditions. It could be that in WT cells you have anisotropic movement due to the radial condensation of the nucleoid and that this anisotropy is lost in A22 treated cells. In these cells mechanisms that operate in WT cells to align the direction of segregation may be disrupted by the loss of cell body symmetry.

As stated by the authors, recent work by Paul Wiggins lab showed that replisomes can split in two. Even earlier work by the Rudner group had already shown that replisomes can split. And even older papers by the Grossman group had shown that cells can display more than 1 replisome. All these data together were already consistent in that a 'pure' replication factory model whereby both replisomes are together was not correct. The data presented in this manuscript confirms these earlier observations.

But, in my opinion the data presented is not necessarily a proof of the train track model. Showing that replisomes can travel apart from each other AND also to come back together would be a step towards this direction.

Other comments

Authors claim that after replisome splitting the fluorescence intensity drops by half but do not display statistics (just point to an image where this is difficult to see). Please provide distribution and number of events.

The authors did all their experiments in strain AB1157. This strain is only exclusively used by the Sherratt lab. Many other labs have observed that this strain displays different chromosome organization and segregation properties than standard E. coli strains. For instance the study of Cass et al Biophys J. 2016 reports major differences between AB1157 and the widely used strain MG1655:

...

Our own live-cell quantitative characterization of the nucleoid in the model strain AB1157 showed that the left-right linear structure was maintained with a very high level of precision and that just 10% of the chromosome (the *ter* region) was decondensed and stretched between the left and the right poles of the nucleoid ([14](<https://www.ncbi.nlm.nih.gov/pmc/articles/PMC4919604/#bib14>)). On the other hand, analysis of MG1655 was consistent with a domain-based organization consisting of four structured macrodomains (*ori*, left, right, and *ter*) and two unstructured regions ([15](<https://www.ncbi.nlm.nih.gov/pmc/articles/PMC4919604/#bib15>)). Although these results were qualitatively consistent with the left-right filament structure, it was difficult to reconcile a *ter* macrodomain in MG1655 with the observation that it was decondensed in AB1157."

...

Thus, it would be important that the authors show the relevance of their results in a widely used strain, not in a strain that is known in the community to be *special*.

Minor Comments

Typo "chromosomes" in Ln 197.

abstract: 'observed chromosome replication with increased width'. Meaning?

Typo line 194 : after which a replication bubble opened → bubble

Point-by-point response to the reviewers' comments

(Reviewers' comments are in black font. Author responses are represented in blue.)

Reviewer 1

This is a beautiful and important study that should and could have appeared ten years in the bacterial chromosome community. The authors employed classic *ts dnaC* mutant to synchronize replication initiation. Furthermore, by treating *E. coli* with a low dosage of A22, cells lose their rod shape and become round, widening their width. This allowed them to clearly visualize the morphology of the fluorescently labeled chromosome and simultaneously track progression of DNA replication from a single intact chromosome using fluorescence microscopy. The end result of this relatively straightforward approach is remarkable.

Before initiation, we see a torus morphology of the circular chromosome. Once replication starts, two replisomes split, unambiguously refuting the factory model. Newly replicated DNA stay cohesed for several minutes but eventually they separate. Near the end of replication, we get to see the figure-8 shape and their separation into two torus. The sequence of these images are worth a figure in a biology textbook, and it is surprising why this type of clear visualization took so long.

We would like to thank the reviewer for the very strong commendation of the work, emphasizing the clarity of our observations of two splitting replisomes, which are “unambiguously refuting the factory model”. We appreciate the enthusiasm of the referee that our “sequence of these images are worth a figure in a biology textbook”.

Perhaps the most striking conclusion of this study is the robustness of chromosome segregation and the role of cell shape in providing the directionality to chromosome segregation. This is similar to the observation by Jeff Errington's group for L-form bacteria. They observed severe defects in chromosome segregation in round cells without cell walls (L-form). However, when the L-form cells were forced to grow in narrow microfluidic channels so that the cells are elongated, chromosome segregation was restored. Errington presented the results in a number of talks at meetings, but somehow I cannot find the reference.

As the authors show that spatial positioning of *ori* (by MukBEF; Ref 29) is not required for chromosome segregation, these observations for the first time provide strong evidence *in vivo* for the entropy-driven segregation models and the role of spatial confinement that has been around for over a decade.

We thank the reviewer for highlighting our finding of the robustness of chromosome segregation and the role of cell shape in providing the directionality to chromosome segregation, as well as the first *in vivo* evidence for entropy-driven segregation. We acknowledge the parallel work of the Errington lab. We have now added a reference to the relevant recent preprint from his lab on L-forms with segregating chromosomes in microfluidics channels (ref. 35).

Overall, the message of the study is simple, clear, and important and should be of very broad interest. The writing is also admirable. In conclusion, I recommend the paper for Nature Communications wholeheartedly.

We thank the reviewer for the strong recommendation to publish the work in Nature Communications.

Minor comment: Typo L194: bible -> bubble

We have corrected the spelling error.

Reviewer 2

The main aim of the study is to discern between two models for replication: the train track and the replication factory models. For this, the authors look at replication dynamics in a strain with a *dnaCts* allele to synchronize cells and therefore follow replication of a single chromosome. In addition, in a second set of experiments, they use conditions under which cells are round-shaped to increase the resolution of detection of the replisomes. The authors set the question as a rebuttal of one of the two existing models for replication. Their results clearly indicate that replisomes can exist as separate entities. This was observed before by other groups (Grossmann, Rudner, Wiggins, Sogaard-Andersen; Berkmen 2006; Wang, 2013; Mangiameli, 2016; Harms, 2013). But I am not convinced that they provide enough evidence for the tracking model.

We would like to thank the reviewer for the thorough assessment of our manuscript. We have added new experimental data that further strengthen our observation that the replisomes are organized as predicted by the 'track model'. We incorporated these new data as an extra figure panel Fig. 2m, which clearly visualizes two replisomes that split and then come back together to the middle of the cell, before the two chromosomes separate.

My first concern is with their observation of nucleoid morphology under their conditions. Nucleoids as observed by HUmYpet labeling looks abnormal: they occupy only a fraction of the cell width and appear abnormally condensed. These abnormalities are usually observed in cells suffering from phototoxic effects (eg. DNA damage) or after deconvolution artifacts.

We understand the concern of the reviewer concerning this important technical point, but there are a number of controls that indicate that labelling and phototoxicity do not pose any problems: First, in an earlier study, we performed extensive controls to make sure that the HUmYpet tagging does not introduce any artefacts in the cell physiology and does not induce phototoxic effects, see our description in (Wu et al. 2015, *Frontiers in Microbiology*). Additionally, our lab extensively characterized why single nucleoids occupy only a fraction of the cell volume while staying physiologically active, see our recent paper (Wu et al. *Current Biology*, 2019). Furthermore, we report that >85% of all cells are able to initiate replication and successfully divide without having growth-related defects at non-permissive temperatures, ruling out significant artefacts due to the photo-toxicity. Finally, we can rule out deconvolution artefacts as we verified our deconvolution imaging with SIM microscopy, which showed the same chromosome morphology as in Fig 4. We took great care to avoid any artefacts arising from the deconvolution of images, as extensively described in (Wu, Japaridze et al, *Nature Comm.* 2019). Taken together, the experimental data show that the cells exhibit a normal physiology and are not stressed, and that the imaging faithfully maps the intrinsic dynamics.

A first conclusion of the ms is that nucleoids display helical patterns. How many cells display this pattern? Other papers claimed similar things (eg. this has been observed before for *Bacillus* (see Ben Yehuda, *PNAS* 2008)) but this was later disproved by other studies (e.g. Marko tested the hypothesis of helicity in their *Mol Micro* paper in 2012, see Yazdi et al) but found no evidence for this. I would thus be aware of any interpretation regarding nucleoid helicity, even more when the nucleoid images appear artifactual (see comment above).

Our observation of helical-like patterns is an interesting observation but not a central point of this paper. We would like to avoid any misconception with the phrasing and therefore now changed the word 'helicity' to 'periodicity'. To measure the periodicity, N=30 nucleoids were chosen at random and the cross-sectional profiles along the long axis were analysed for autocorrelation. The number of analysed cells is indicated on panel fig.2e as well as in the caption. If requested we can provide all individual height cross-sections and individual autocorrelation graphs in SI.

One main result is that 1 replisome focus appears at midcell, then splits and moves away. This is interpreted to be in support of the train track model. But in this model one would also expect the

replisomes to come back together and fuse at midcell as they turn to the replication of the terminus. Why is this not observed?

We agree with the scenario sketched by the referee, namely that the train track model would predict that the replisomes will come together again after their initial splitting. We now performed additional experiments to monitor whether this indeed is happening. And it is. We now include these new data in panel Fig.2m. This observation that the replisomes are coming back together is clear further evidence that is strengthening the support for the train track model.

The experiment presented in Fig 1 has already been realized in Mangiameli et al (Plos Genetics 2016) under seemingly very similar conditions. In that case, the main conclusion was that replisomes remain close to midcell. Why are the results so different?

Indeed, as the referee correctly points out, the conclusions drawn from our results and by Mangiameli et al are very different. Different experimental conditions were used by both groups however. The main advantage of our experimental conditions, as described in the paper, is the use of a temperature-sensitive strains with elongated cell shapes. In our experiments, the cells are grown for several hours under permissive temperatures so that their chromosomes can elongate within the available cytosolic space. This leads to cell and chromosome sizes that are much longer than the typical cell sizes observed by Mangiameli et al., which permitted us to resolve the splitting of the replisomes. A second main difference between our work and that of Mangiameli et al, is the way that the beta clamps (DNA-N) were tagged. As discussed in our manuscript, we used monomeric mCherry tag for the beta clamp, in order to prevent any artefacts related to fluorophore-fluorophore interactions, while non-monomeric YPet or GFP proteins were used by Mangiameli et al.

The authors explain that the number of cells with 2 replisome foci decreases (drastically) from 70 to 35% after 30' because of photobleaching or replisome disassembly. But photobleaching should be a gradual effect, not drastic (this can be actually measured by following the focus intensity over time...). But why would the replisomes disassemble before they replicate the ter?? In whatever case it seems to me that they must fuse before. Otherwise, it could be that disassembly is due to phototoxic effects.

Apparently, our text was unclear, as we did not at all intend to say that the replisomes disassemble before finishing the replication. We now updated the text on page 5 to avoid such a misconception.

We indeed mentioned a reduction of the number of replisome spots observed at a time point 30 minutes after replication initiation. As in these measurements, we cannot distinguish replisome bleaching from a disassembling of the replisome after finishing the replication, we now mention both effects.

Triggered by the comment of the referee, we collected new data, now added as Supplementary Fig. S6, that show the gradual bleaching of the replisomes. Furthermore, we show that single replisome spots, that presumably contain two replisomes, have a roughly twice higher integrated fluorescence intensity compared to the spots that are observed in cells where two replisomes were spatially resolved in separate foci.

The experiment in Figure 2 k,I is key to the author's claim that they observe replisome tracking. However a single example is shown. Authors should perform analysis based on a statistically relevant number of cells where they show single-cell traces of the distance between replisomes as a function of time. This is essential to support their claims. Statistics of number of replication spots from distinct cells at different time intervals (Figure S3) do not provide any support for a tracking model as both tracking and factory models would yield similar results.

We agree with the reviewer that proper statistics should be done on a high number of cells, and this was indeed also done in our manuscript. The data reported in Figure 2 and 3 and the supplementary video2, are typical representations of the general behaviour. The splitting of the foci was observed in more than 70% of N=147 cells, as mentioned in our paper on page 5.

As to the last sentence of this comment, we disagree with the reviewer on the point that the track model and factory model would yield the same number of foci during replication. The factory model predicts a single replisome spot throughout the entire replication process, whereas the track model predicts two spots throughout most of the cell cycle, as observed in our experiments.

In Fig. 3 the authors use A22 to induce changes in cell morphology. These lead to expansion of the cell volume and the decompaction of the nucleoid with respect to the conformation of the nucleoid in normal cells. It is an interesting method but I wonder whether the behaviour of replisomes and segregation is relevant to wildtype conditions given these large changes in chromosome compaction and organization.

Widening replicating cells with low amounts of A22 does not change the growth and the segregation of the chromosomes, as reported by the lab of Sherratt and others, as well as by our own lab (see Wu, Japaridze et al, Nature Communications 2019). Notably, we observed replisome splitting in these rounded cells as well as in elongated rod-shape cells, which clearly indicates that the replisome splitting is not due to the widening of the cell.

However, if the cells have a single chromosome, the segregation process is indeed affected by the cell width, as is discussed in Figure 4.

The data in Fig. 3b are very unclear. The authors seem to show DNA images annotated with a circle that seems to indicate the positions of replisomes. Are the positions of these circles based on images of DnaN as in Figure 2? If it is the case, please show raw data and merged images. Otherwise, I just don't understand how they can get the positions or replisome foci based purely on DNA signal.

As requested by the reviewer, we have now added the raw images of the replisomes to Figure 3.

I cannot really understand how the authors can correlate the schemes in 3a to the images/traces in 3c-d. In the schemes they see the appearance of a replication bubble that increases in size as replication progresses. However, in the images below we observe that replisomes do not appear at the regions of high DNA density but more often at their borders (panels I-III) or even far away (panel IV). Also, it is not clear in panel IV what this high density regions are as they do not seem to be the regions where DNA is being replicated.

The reviewer is correct that the positions of the replisomes do not strictly coincide with the high DNA intensity regions in each and every image. However, the correlation analysis, described in Supplementary Fig.S8, shows that the replisome spots in fact do reside predominantly in the vicinity (that is: within or very close to it) of high-density DNA clusters that contain twice more DNA material compared to random DNA markers. We improved the clarity of the presentation of the data by updating panel Fig.3b.

Then the authors claim they see sister chromatid cohesion, however they just show an image where they speculate a shade in the fluorescence signal may correspond to the replication bubble and where more intense DNA regions may correspond to cohesed sisters. But this is not evidence for sister chromatic cohesion. The authors should provide experimental proof for this claim.

Sister cohesion has been reported previously by several other groups, to which we also refer in the manuscript. As we observed that the local chromosome segregation was delayed as compared to the passage of the replisome spots, and furthermore observed a delayed sister chromosome

separation in experiments with Novobiocin, we think it is useful to point out to the reader that these observations are consistent to the sister chromatid cohesion that was reported before by others. However, sister cohesion is not the main point of our paper, and therefore we more accurately described our point and softened the in the main text.

The authors show in 3e that A22-treated cells are able to complete chromosome replication and segregation. This is good to see. But it would be important to show where the replisomes are during this process. Why don't they show the replisomes images? You would expect the replisomes coming together to the link in the figure-8. This could be an important support of their results.

As mentioned in the manuscript, bleaching of the fluorophores at the replisomes was a major bottleneck in such timelapse experiments. While we succeeded in tracking the replisomes in rod-shaped cells over the full cycle in new experiments (as discussed above), it has turned out too hard to image the replisomes throughout the entire replication cycle in the A22-treated cells.

The authors then use a strain with a FROS at oriC and a second close to ter and image how these FROS organize in cells treated with A22. A main conclusion, that is different from previous studies, is that newly replicated origins move away from each other isotropically. But the experiments in this paper are performed with this widened cells where cytoplasm volume is much larger, therefore I do not see the relevance to wildtype conditions. It could be that in WT cells you have anisotropic movement due to the radial condensation of the nucleoid and that this anisotropy is lost in A22 treated cells. In these cells mechanisms that operate in WT cells to align the direction of segregation may be disrupted by the loss of cell body symmetry.

It is good to read that the reviewer shares our point of view on this observation. Indeed, as pointed out in the discussion (p. 9), we also conclude that the initial isotropic movement of the ori's is due to the loss of the radial confinement.

As stated by the authors, recent work by Paul Wiggins lab showed that replisomes can split in two. Even earlier work by the Rudner group had already shown that replisomes can split. And even older papers by the Grossman group had shown that cells can display more than 1 replisome. All these data together were already consistent in that a 'pure' replication factory model whereby both replisomes are together was not correct. The data presented in this manuscript confirms these earlier observations.

We agree with the reviewer that our work is consistent with early works by Rudner, Grossman and colleagues that indicating that the replisomes are not organised into a factory model. However, we note that the controversy persists until today, as shown by recent papers from e.g. the Merrikh and Wiggins labs that still advocate a factory model, see Mangiameli et al. Plos One (2017); Mangiameli et al. Current Genetics (2018). Our data now provide new, and in our view compelling, evidence for the track model.

But, in my opinion the data presented is not necessarily a proof of the train track model. Showing that replisomes can travel apart from each other AND also to come back together would be a step towards this direction.

We hope that with the additional experimental evidence that has been added to the manuscript, as discussed above, we now convinced the referee that the replisomes are indeed organised according to the track model.

Other comments

Authors claim that after replisome splitting the fluorescence intensity drops by half but do not display statistics (just point to an image where this is difficult to see). Please provide distribution and number of events.

As requested by the reviewer, we have now added the statistical number of the events in the manuscript.

The authors did all their experiments in strain AB1157. This strain is only exclusively used by the Sherratt lab. Many other labs have observed that this strain displays different chromosome organization and segregation properties than standard *E. coli* strains. For instance the study of Cass et al Biophys J. 2016 reports major differences between AB1157 and the widely used strain MG1655: "...Our own live-cell quantitative characterization of the nucleoid in the model strain AB1157 showed that the left-right linear structure was maintained with a very high level of precision and that just 10% of the chromosome (the *ter* region) was decondensed and stretched between the left and the right poles of the nucleoid ([14](https://www.ncbi.nlm.nih.gov/pmc/articles/PMC4919604/#bib14)). On the other hand, analysis of MG1655 was consistent with a domain-based organization consisting of four structured macrodomains (*ori*, left, right, and *ter*) and two unstructured regions ([15](https://www.ncbi.nlm.nih.gov/pmc/articles/PMC4919604/#bib15)). Although these results were qualitatively consistent with the left-right filament structure, it was difficult to reconcile a *ter* macrodomain in MG1655 with the observation that it was decondensed in AB1157."

Thus, it would be important that the authors show the relevance of their results in a widely used strain, not in a strain that is known in the community to be *special*.

Although earlier reports indicated that some finer details of the chromosome organization in the MG1655 and AB1157 strains, such as the macrodomain organization, may be slightly different, we compared these two strains and found that there were no significant differences between them as the global circular chromosome organization was preserved in both strains (see Wu, Japaridze et al, 2019) (and hence we would hesitate to call any of the strains 'special'). Particularly, we would not expect a difference in the replication behavior between both strains as e.g. the *ori* was shown to be similarly organized in both strains. We feel that an inter/cross-species comparison is somewhat redundant and beyond the scope of the work (while it amounts to a very significant amount of additional experimental work). In the conclusions section on page 9, we now specified explicitly that our data were collected in the AB1157 strain of *E. coli*.

Minor Comments

Typo "chromomes" in ln 197.

The spelling error was corrected

abstract: 'observed chromosome replication with increased width'. Meaning?

We rephrased the sentence in the abstract to avoid unclarity.

Typo line 194 : after which a replication bible opened → bubble

The spelling error was corrected

Reviewers' comments:

Reviewer #2 (Remarks to the Author):

Many of my original concerns were answered by the revision and the point-by-point answers. However, there are still some key points that remain to be solved (see below).

1- nucleoid morphology. The original question raised concerned the fact that nucleoids in this study occupy only a fraction of the cell (appearing artifactually shrunked), while other studies from many different labs showed that this is not the case (e.g. Stracy PNAS 2015; Makela & Sherratt, Mol Cell 2020; Le Gall, Methods in Mol Biol. 2016;). The authors's answer is that in their previous studies they observed the same shrunked morphologies, but this is not what we see in their cited articles. For instance, in Wu et al. Frontiers in Microbiology 2015 the nucleoids look normal, unlike shrunked nucleoids in the present study. This is an important question that still needs to be explained. For example the authors could acquire nucleoid images of other background strains (eg MG1655) to ensure this is not an artifact of their imaging conditions in the current study or from the temperature-sensitive strains used here.

2- observation of replisomes coming back to midcell. Their new data in Figure 2m is very welcoming as this is a key observation to support their model. However, a single cell is shown in Fig. 2m. As in the original submission, there are no number of cells in which this event has been observed or the number of biological replicates (even when this is typically requested by NPG journals). There is neither any sort of quantitative analysis based on this imaging. Thus, while it is reassuring to see one instance where replisomes come together, this does not amount to evidence. The authors do not report replisomes coming back together in A22 treated cells, thus, this is the only evidence they present in the current revision.

Point-by-point response to the reviewers' comments

(Reviewers' comments are in black font. Author responses are represented in blue.)

Reviewer 2

Many of my original concerns were answered by the revision and the point-by-point answers.

We thank the reviewer for noting our efforts to answer all of his/her questions.

However, there are still some key points that remain to be solved (see below).

1- nucleoid morphology. The original question raised concerned the fact that nucleoids in this study occupy only a fraction of the cell (appearing artifactually shrunked), while other studies from many different labs showed that this is not the case (e.g. Stracy PNAS 2015; Makela & Sherratt, Mol Cell 2020; Le Gall, Methods in Mol Biol. 2016;). The authors's answer is that in their previous studies they observed the same shrunked morphologies, but this is not what we see in their cited articles. For instance, in Wu et al. Frontiers in Microbiology 2015 the nucleoids look normal, unlike shrunked nucleoids in the present study. This is an important question that still needs to be explained. For example the authors could acquire nucleoid images of other background strains (eg MG1655) to ensure this is not an artifact of their imaging conditions in the current study or from the temperature-sensitive strains used here.

Triggered by this comment of the referee, we collected new data that show images of nucleoids of the MG1655 and AB1157 strains without replisome labels or FROS arrays. As suggested by the reviewer, we do indeed observe some differences between the morphology of the nucleoids between these two *E. coli* strains. While noteworthy, this does not relate to the main message of the current paper. And furthermore, the overall scaling behavior is remarkable similar for both strains. We think these new data on the comparison between strains are a useful addition to the paper and we have therefore now added these as Supplementary Fig. S3.

In passing we like to mention that we disagree with the suggestion that the nucleoids 'appear artefactually shrunked'. In the studies mentioned by the reviewer, Makela & Sherratt (Mol Cell. 2020) did not show any images of the nucleoids (as they imaged MukB, ori, and ter signals, but not the nucleoids), while in both the works by Stracy (PNAS, 2015) and Le Gall (Nature Comm. 2016), the authors used the MG1655 strain, and the slight difference in chromosome morphology could, as indicated above, be the result of that.

2- observation of replisomes coming back to midcell. Their new data in Figure 2m is very welcoming as this is a key observation to support their model. However, a single cell is shown in Fig. 2m. As in the original submission, there are no number of cells in which this event has been observed or the number of biological replicates (even when this is typically requested by NPG journals). There is neither any sort of quantitative analysis based on this imaging. Thus, while it is reassuring to see one instance where replisomes come together, this does not amount to evidence. The authors do not report replisomes coming back together in A22 treated cells, thus, this is the only evidence they present in the current revision.

As requested by the reviewer, we now added both requested items:

(i) We added the relative number of replisomes coming back toward each other in the main text, as well as indicate that all experiments were repeated at least in duplicates in the materials and methods section.

(i) We added an example of images of a A22-widened cell where the replisomes move along the chromosome and then come back together, see the new Figure 3f.

REVIEWERS' COMMENTS:

Reviewer #2 (Remarks to the Author):

The authors have addressed the main issues I had:

- lack of number of cells in Fig. 2m
- example of replisomes rejoining in lemon-shaped cells.
- nucleoid morphologies in their strain being abnormal.

I am satisfied with these additions, as they answer some of my previous concerns.

I would only request that:

- the authors systematically add cell numbers for all their panels. For instance this is lacking for their newly added data in Fig 3f and maybe in others I missed. This is standard policy for NPG.
- From the new data presented, it is now clear that nucleoids of AB1157 are considerably more condensed than typically observed for the widely used wildtype EC strain MG1655. I have raised in a previous review the controversies surrounding this strain (AB1157) reported in the literature, and many other known differences with MG1655 that are known in the community but usually not reported. Thus, I think it is reasonable that the authors discuss this in the Discussion section by explicitly mentioning the differences in nucleoid morphologies/segregation of Ter/ etc between these strains. This is important as they could be at the origin between the model reported here and that proposed previously in the literature.

Point-by-point response to the reviewers' comments

(Reviewers' comments are in black font. Author responses are represented in blue.)

Reviewer 2

The authors have addressed the main issues I had:

- lack of number of cells in Fig. 2m
- example of replisomes rejoining in lemon-shaped cells.
- nucleoid morphologies in their strain being abnormal.

I am satisfied with these additions, as they answer some of my previous concerns.

We are happy to hear that the reviewer is satisfied with additional data presented in the revised manuscript and with our answers to his/her questions.

I would only request that:

- the authors systematically add cell numbers for all their panels. For instance this is lacking for their newly added data in Fig 3f and maybe in others I missed. This is standard policy for NPG.

We added the relevant statistics in the missing figure captions.

- From the new data presented, it is now clear that nucleoids of AB1157 are considerably more condensed than typically observed for the widely used wildtype EC strain MG1655. I have raised in a previous review the controversies surrounding this strain (AB1157) reported in the literature, and many other known differences with MG1655 that are known in the community but usually not reported. Thus, I think it is reasonable that the authors discuss this in the Discussion section by explicitly mentioning the differences in nucleoid morphologies/segregation of Ter/ etc between these strains. This is important as they could be at the origin between the model reported here and that proposed previously in the literature

As requested by the reviewer we added these points to the manuscript.